# Cognitive Gain in Digital Foreign Language Learning

**DOI:** 10.3390/brainsci13071074

**Published:** 2023-07-15

**Authors:** Blanka Klimova, Marcel Pikhart

**Affiliations:** Department of Applied Linguistics, Faculty of Informatics and Management, University of Hradec Kralove, 500 03 Hradec Kralove, Czech Republic; marcel.pikhart@uhk.cz

**Keywords:** cognitive improvement, cognitive benefits, foreign language learning, second language

## Abstract

This systematic review examines the potential of digital language learning in contributing to students’ cognitive gains. The study reviews existing research on the relationship between digital language learning and cognitive benefits, with a focus on enhanced problem-solving skills, memory, and multitasking ability. The research questions explored in this study are (1) does digital language learning contribute to cognitive gains in foreign language education? and (2) what are the pedagogical implications for cognitive improvement in digital foreign language education? The study employs the Preferred Reporting Items for Systematic Reviews and Meta-Analyses (PRISMA) methodology to identify and analyze relevant research articles. The results of the review suggest that working with printed texts may be more effective for cognitive gains compared to electronic texts. Additionally, implementing more senses through digital language education appears to be beneficial for cognitive gains. Thus, several pedagogical implications emerge for promoting cognitive improvement in digital foreign language education. Firstly, it is crucial to implement techniques and strategies that best align with students’ language needs in a digital learning environment, whether it involves pen-and-paper activities or a flipped classroom approach. Secondly, exposing students to a variety of techniques that engage multiple senses can have a positive impact on cognitive gains. Finally, providing students with feedback is essential to maintain their motivation and foster continued progress in their foreign language studies.

## 1. Introduction

Research [1,2] shows that learning a foreign language can significantly contribute to cognitive gains, leading to benefits such as enhanced problem-solving skills, memory, and multitasking ability, irrespective of age [3]. The findings of research studies also reveal that digital competences positively correlate with language proficiency outcomes. Participants with higher levels of digital competences achieve better language learning outcomes compared to those with lower levels of digital competences (cf. [4]). Furthermore, digital technologies can offer interactive and engaging methods of learning, which can positively affect retention and understanding more than traditional methods [5,6,7]. This is especially true for the application of digital games in foreign language learning. Peterson et al. [8] state that gameplay enhances peer collaboration, the production of target language output, vocabulary learning, and reduces the influence of factors that inhibit learning. This was confirmed by another study by Klimova and Kacetl [9], who claim that computer games positively affect vocabulary acquisition in foreign language learning, as well as provide exposure to the target language, increased engagement, and enhancement of learners’ involvement in communication.

As indicated above, digital technology can have a positive effect on the development of cognitive skills at any age. A research study conducted by Di Giacomo, Ranieri, and Lacasa [10] among 191 children aged between 7 and 10 years shows that their exposition digital gaming improved their learning capability not only in terms of high success in the educational path but much more in the cognitive functionality by enhancing the verbal/visuoperceptual performance. Similar results were also found among older individuals for whom computer-based foreign language training programs may bring cognitive benefits, especially as far as the enhancement of their cognitive functions, such as working memory, is concerned (cf. [11]).

Moreover, implementing digital technologies into foreign language teaching and learning nowadays appears to be a popular method of language acquisition [12] since technologies can provide easy and fast access to diverse linguistic resources, teaching tools, and community interactions. They also enable learners to proceed at their own pace from anywhere and at any time and this makes learning more personalized [13] and self-directed [14], which consequently leads to more effective learning [15].

Overall, the implementation of digital technologies in foreign language teaching and learning offers a range of benefits, including access to diverse resources, interactive teaching tools, community interactions, personalized and self-directed learning, and enhanced learning effectiveness [16]. These advantages have made digital technologies a popular method for language acquisition in today’s educational landscape.

However, digital foreign language learning also has some limitations. The lack of face-to-face interaction can result in missing consolidating some of the language linguistic phenomena that are drilled in face-to-face classes [17]. Additionally, the self-directed nature of digital learning requires a high level of motivation and discipline, which can pose challenges for some learners [18]. Research also indicates that technology does not bring any benefit for cognitive gains, which was true, for example, for a Finnish study [19] that investigated the use of information and communications technology (ICT) at secondary schools among 5037 students. Their results showed that frequent ICT use at schools was associated with students’ weaker performance in all the cognitive learning outcomes.

Acknowledging these limitations can help educators and learners make informed decisions about the appropriate use of digital technologies in foreign language education. A balanced approach that combines digital resources with face-to-face interactions and effective pedagogical strategies can help overcome some of these limitations and provide a comprehensive language learning experience.

This review aims to explore the impact of digital language learning on students’ cognitive development. To achieve this, the following research questions have been formulated:Does digital language learning contribute to cognitive gains in foreign language education?What are the pedagogical implications for cognitive improvement in digital foreign language education?

## 2. Materials and Methods

The analysis in this study employed the Preferred Reporting Items for Systematic Reviews and Meta-Analyses (PRISMA) [20] procedure to retrieve relevant studies. The search process consisted of four phases: identification of relevant research articles, their screening, eligibility evaluation, and inclusion in the review. During the identification phase, existing research on the possibilities of cognitive gain in digital foreign language learning and related terms in the literature were reviewed to compile a list of keywords for the article search. Screening involved assessing the accessed papers based on predefined inclusion/exclusion criteria, considering factors such as review timeline and publication type (e.g., journal articles, conference proceedings, book reviews). Eligibility entailed evaluating the identified records in relation to the research questions and study objectives, leading to final decisions regarding their inclusion in the review.

In order to determine the search terms, the research team conducted online sessions and engaged in discussions to develop a consensus. These discussions resulted in the creation of two sets of keywords based on the topic, i.e., a combination of various aspects related to cognitive gain and foreign language learning using digital tools.

The first set of search terms included “cognitive” combined with “gain”, “advantage”, and “benefit”. The second set encompassed terms combining “language” with words like “education”, “acquisition”, “teaching”, “learning”, “foreign”, and “second” to focus the search within the field of language education. These two sets were combined using Boolean operators (i.e., OR and AND) to capture various combinations of these key terms.

The search string was as follows:


**
*(“cognitive gain *” OR “cognitive improvement *” OR “cognitive benefit *”) AND (language AND (education OR acquisition OR teaching OR learning OR foreign OR second))*
**


Whether the research focused on digital tools used for language learning, it was then evaluated by the research team if to include the article in the review process. Detailed inclusion and exclusion criteria are as follows.


**
*Inclusion criteria*
**


Journal articles in the English language;Empirical research articles;EFL/ESL/L2 context;Articles published after the year 2000;Foreign language learning supported by any kind of digital tool;Open access articles.


**
*Exclusion criteria*
**


Conference proceedings, book chapters, editorials;Other languages;Descriptive studies, theoretical studies, reviews, commentaries;Other context than digital foreign language learning.

To access published literature on cognitive gain in digital foreign language learning, the research team searched two established and authoritative databases commonly used, namely, Web of Science, and Scopus. After performing an automated search through the databases, a manual search approach was employed to ensure comprehensive results and maximize the initial pool of records. In this regard, the literature search also incorporated referential backtracking, which involved examining the references of relevant documents to ensure no relevant papers were overlooked.

It is also obvious that the researchers tried to implement as many articles as possible during the screening process; however, the major issue was that there were not many purely empirical studies which led to massive removal of the majority of articles generated by the databases. Out of the initial search of 256 articles, the research team carefully evaluated each one, considering the inclusion and exclusion criteria. As a result of this thorough process, nine studies were identified that perfectly met all the criteria, making them suitable for inclusion in the review as is seen in Figure 1.

## 3. Results

Altogether, nine studies were generated through the database and reference searches. They originated across all continents. The timeline of their origin spans from 2013 to 2023 inclusive. The main topic of all these studies focuses on cognitive gains/benefits/improvements in digital foreign language education. The research sample ranges from 17 to 122 subjects. The outcome measures vary according to different research designs that required the nature of the authors’ research. Nevertheless, all studies employed standard evaluation methods, such as pre- and post-tests, questionnaires, or interviews, as well as statistical processing.

The results of these studies generate two important outcomes. Firstly, it seems that working with printed texts is more effective for cognitive gains than with electronic ones, as shown in the research findings by [21,22,23]. Secondly, implementing more senses with the help of digital foreign language education seems to be beneficial for cognitive gains when learning a foreign language (cf. [5,6,7,14,24,25]).

Table 1 below provides more specific details of the key findings of the detected studies. The findings are described in alphabetical order by the first writer of the research study.

## 4. Discussion

The results of this review study described above show that the findings about cognitive gains/improvement in digital foreign language learning vary. Thus, the answer to the first research question of whether *digital language learning contributes to cognitive gains in foreign language education* or not is inconclusive since digital foreign language learning has its cons and pros depending on a number of variables, such as a method used for learning a foreign language, students’ learning needs, motivation or a digital tool employed. On the one hand, it appears that pen-and-paper strategy for the retention of new words is still more beneficial than using a language application. As Mangen, Anda, Oxborough, and Brřnnick [22] report, writing with a ballpoint pen is sensorimotorically and kinesthetically different from the process of writing by tapping keys on a keyboard. Umejima, Ibaraki, Yamazaki, and Sakai [26] also claim that handwriting promotes the acquisition of rich encoding information and/or spatial information of real papers. In addition, Pikhart, Klimova, and Ruschel [23] maintain that students need to make notes of new words and phrases, highlight them or write their translation in their native language to retain them even after a longer period. This is also true for other printed materials, such as printed dictionaries, which were used in a research study by [21]. Their findings reveal that it is the fast retrieval speed of electronic dictionaries that might hinder the retention of new words.

In summary, the relationship between digital foreign language learning and cognitive gains is not definitive, as indicated by the findings of this review. It must be highlighted that the effectiveness of digital language learning in enhancing cognition depends on factors such as the learning approach, individual needs, motivation, and the specific digital tools used. It is important to note that various activities that are analogous, such as taking notes, highlighting, or writing translations in one’s native language, can significantly improve long-term retention of vocabulary [23]. Similarly, printed dictionaries have demonstrated greater efficacy than electronic ones, as the quick retrieval speed of electronic resources may impede word retention [21].

On the other hand, it seems that it depends on the techniques/strategies that are employed for learning a foreign language. For example, digital games seem to be beneficial as the findings by Bancha and Tongtep [14] reveal. This is in line with a study by Vnucko and Klimova [27], who state that digital games create an environment in which students experience predominantly positive emotions which might then contribute to students’ retention of new words. Furthermore, Huang, Willson, and Eslami [28] report that games not only stimulate students to learn but also enable them to pay longer attention to vocabulary lessons. Generally, as Park [5] claims, the more senses are applied in a foreign language, the better since students’ retention of new vocabulary increases. Another approach, which is quite widespread at the moment and especially after the COVID-19 pandemic, is the use of a flipped classroom or inverted classroom, which provides more time for the teacher in class to focus on those linguistic phenomena that students need to acquire to become more proficient users of the target language on the condition that foreign language teachers provide their students with sufficient support in an online environment [29]. The findings of this review reveal that the flipped classroom approach seems to bring cognitive improvement in both writing [6] and speaking skills [7]. Öztürk and Çakıroğlu [30] also point out that implementing a flipped classroom approach can contribute to the enhancement of self-regulated strategies among foreign language learners.

The core message that must be stressed here is that the effectiveness of digital foreign language learning depends on the techniques and strategies employed. Different approaches, such as digital games and the flipped classroom, have shown potential benefits for cognitive improvement in language learning. Digital games create a positive emotional environment and stimulate students’ attention and retention of new words. The use of multiple senses in language learning, as well as providing support in an online environment, can enhance vocabulary retention. The flipped classroom approach, when supported adequately, has demonstrated cognitive improvements in both writing and speaking skills.

As far as the second research question on *pedagogical implications for cognitive improvement in digital foreign language education* is concerned, the results of this review suggest the following:Implementing such a technique or a strategy that would best suit students’ language needs in a digital foreign language environment, be it pen and paper or flipped classroom;Exposing students not only to one technique but to several that would activate most of their senses that have a positive impact on cognitive gains;Providing students with feedback in order to keep them motivated to continue in their foreign language studies [31,32].

Utilizing an appropriate technique or strategy in a digital foreign language setting, such as pen-and-paper activities or a flipped classroom approach, tailored to meet the individual language needs of students will ground students in a real setting so that they will not be disconnected from reality. Moreover, exposing students to multiple techniques rather than relying on a single approach, ensuring the engagement of various senses, will maximize potential cognitive benefits that are related to the use of technologies. And finally, supplying students with feedback seems crucial to maintaining their motivation and encouraging their foreign language learning endeavors. However, such approaches require teachers to be well equipped with digital pedagogy. Therefore, Aldhafeerin and Alotaibi [33] emphasize holding workshops for all education stakeholders to educate them about digital pedagogy in foreign language education since the teacher’s role in implementing technology into the learning processes is of paramount importance and such support is highly beneficial [12].

The limitations of this review reflect a relatively smaller amount of research studies on the topic. However, the findings of these studies are quite representative since they include different regions, as well as various digital tools employed in foreign language learning in order to achieve cognitive gains.

Although the availability of empirical research is limited, it still offers a preliminary basis for future studies. Additionally, this study can serve as a catalyst for further research, given the widespread use of digital tools in learning, making it crucial to explore their potential impact on cognitive development.

Future lines of research, following the findings of this study, could focus on the analysis of the possibilities of various AI-driven tools to enhance cognitive skills so that they can be used in various contexts, such as educational, societal, and others. However, it will also be necessary to empirically verify not only the possibilities but also various threats that could arise from the use of AI tools from various perspectives, such as addiction, loss of social contacts, isolation, challenges related to psycholinguistics and language development, and many others. The research into these areas is necessary and, unfortunately, missing. Verification is needed as to where AI-driven tools could cognitively stimulate and where they present a serious threat to our cognitive social and psychological development. None of these questions have been raised nor answered yet. Therefore, this review is a call for these endeavors that seem needed in the global digital world where human–computer interaction is a truism in, literary, all aspects of human life.

## 5. Conclusions

In conclusion, the findings of this review highlight the variability in cognitive gains observed in digital foreign language learning. The question of whether digital language learning contributes to cognitive improvement in foreign language education remains inconclusive due to the pros and cons associated with different variables, such as learning methods, student needs, motivation, and digital tools used. While pen-and-paper strategies appear to be more beneficial for word retention compared to language applications, there is evidence supporting the positive impact of digital games on vocabulary acquisition. The flipped classroom approach also shows promise in enhancing cognitive abilities in writing and speaking skills.

Based on the results of this review, several pedagogical implications emerge for promoting cognitive improvement in digital foreign language education. Firstly, it is crucial to implement techniques and strategies that best align with students’ language needs in a digital learning environment, whether it involves pen-and-paper activities or a flipped classroom approach. Secondly, exposing students to a variety of techniques that engage multiple senses can have a positive impact on cognitive gains. Finally, providing students with feedback is essential to maintain their motivation and foster continued progress in their foreign language studies.

Although this review has identified certain limitations, such as the limited availability of research studies in this area, the findings still offer a preliminary foundation for future investigations. In addition, the authors consider this review important since so far there has not been any review on this research topic. and that is why the findings fill in the gap in the existing literature on cognitive gains in digital foreign language education. As digital tools continue to play a significant role in learning, further research is needed to explore their potential impact on cognitive development. Overall, this study serves as a starting point for future inquiries and emphasizes the need for continued exploration of digital foreign language education and its implications for cognitive enhancement.

## Figures and Tables

**Figure 1 brainsci-13-01074-f001:**
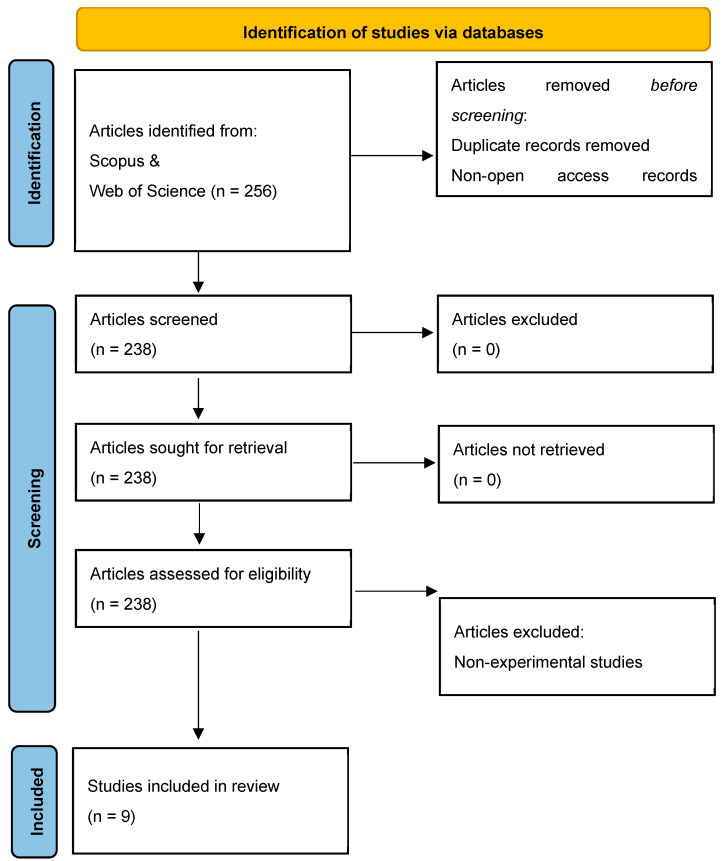
PRISMA flow chart: identification of studies.

**Table 1 brainsci-13-01074-t001:** An overview of the findings of the detected studies.

Study	Objective	Research Sample and Procedure	Outcome Measures	Findings
Bancha and Tongtep [14] Thailand	To explore whether vocabulary lessons plus LMS exercises and vocabulary lessons plus MultiEx games enhance short-term vocabulary memorization and long-term vocabulary retention.	A total of 72 first-year Thai students (37 females and 35 males); with a low level of English; divided into two study groups: face-to-face lessons + LMS exercise and face-to-face lessons + MultiEx Game groups learning new English words, which were based on the Test of English for International Communication (TOEIC). This lasted 10 weeks.	Pre-test, immediate post-test, delayed post-test, statistical processing.	The results showed a higher mean score for the MultiEx game group in both the immediate post-tests and the delayed post-test. However, both groups improved their language proficiency.
Chiu and Liu [21] Taiwan	To examine the impact of using printed dictionaries (PD), pocket electronic dictionaries (PED), and online type-in dictionaries (OTID) on English vocabulary retention among junior high school students.	A total of 33 seventh graders (19 males and 14 females) were asked to use all three types of dictionaries to finish reading tasks (three articles, each 200 words long, adapted from a widely used English learning magazine published in Taiwan).	Pre-test, immediate post-tests, delayed post-test of vocabulary, questionnaires, and interviews.	Results indicate that although electronic dictionaries (OTID and PED) temporarily attract junior high school students’ attention, PD helps them retain target words more effectively.
Liao, Kruger, and Doherty [25] Australia	To investigate the impact of bilingual subtitles on cognitive load and comprehension in L2.	A total of 20 Chinese native postgraduate program students who used English as their second language (L2); fourteen females and six males; each saw four videos only, i.e., either English narration with Chinese subtitles (CS); or English narration with English subtitles (ES); or English narration with both Chinese and English subtitles (BS); or English narration without subtitles (NS).	Biographical questionnaire, cognitive load questionnaire, eye tracking, statistical processing.	Bilingual subtitles are more beneficial when compared with no subtitles as they provide linguistic support to make the video easier to comprehend and facilitate the learning process. However, they do not result in cognitive overload and impede comprehension as a result of increased redundancy.
Mangen, Anda, Oxborough, and Brřnnick [22] Norway	To explore the impact of writing modality on word recall and recognition.	A total of 36 females aged 19–54 years; they were college students or staff at a middle-sized Norwegian university; participants were required to use handwriting, a physical laptop keyboard, and an iPad virtual touch keyboard, each to write down a different word list.	Word list paradigm, statistical processing.	The results were significantly better for free recall of words written in the handwriting condition, compared to both keyboard writing conditions.
Park [5] China	To investigate the impact of a technology-enhanced real-world environment on foreign vocabulary acquisition.	A total of 48 adult participants of 20 different nationalities, living in Newcastle, UK (aged between 19 to 49 years; 16 males and 32 females) had two cooking sessions: one in a kitchen using real objects and the other in a classroom looking at photos. All were learning basic Korean.	Pre-test, immediate post-test, delayed post-test, statistical processing.	Engaging all senses in a technology-enhanced environment is more powerful for vocabulary learning than using only a few senses.
Pikhart, Klimova, and Ruschel [23] Czech Republic, Brazil	To evaluate vocabulary retention in L2 when using print text in contrast with digital media.	A total of 122 university students (66 males and 56 females); divided into two groups to learn 60 new phrasal verbs in 4 weeks; one group of them used a standard print text, and the other used the same text displayed and annotated on their digital devices. The level of students’ English was B2 to C1 according to the Common European Reference Framework for languages.	Vocabulary memory post-tests; statistical processing.	The findings indicate that students using the print text performed better in both tests.
Roussel and Galan [24] France	To examine the effect of clicker use to support learning in a dual-focused second language German course.	A total of 36 Law School students and German language learners (Levels B1–C1); students answered questions with or without clicker use during the 10-week semester. Five sessions were conducted with clickers and five sessions without.	Pre-test and post-test questionnaire, statistical processing.	The clicker group outperformed the non-clicker group with regard to a post-test concerning legal terminology. The findings illustrate that clicker use enhances the cognitive load induced by learning both new terminology and content.
Shafiee Rad, Roohani, and Rahimi Domakani [6] Iran	To assess the effectiveness of a technology-enhanced flipped classroom on English language learners’ expository writing skills.	A total of 60 female students of English as a foreign language, aged between 23 to 38 years; all had an advanced level of English; they were divided into the control group (17 students) and experimental groups: discussion-oriented flipped group/classroom (*n* = 19) and role-reversal flipped group/classroom (*n* = 24).	English placement test, pre-test, and post-test, questionnaire, semi-structured interviews, and statistical processing.	The experimental groups were more effective than the control group in writing gains. In addition, the role-reversal group outperformed the discussion-oriented group in the writing gains in the post-test expository essays.
Yang, Yin, and Wang [7] USA	To explore the application of the flipped learning approach in the instruction of Chinese as a foreign language.	Two first-year Chinese classes; one traditional, face-to-face class (nine students) and the other flipped (online) class (eight students) learning basic Chinese during one semester.	Oxford Language Aptitude Test, Chinese placement test, final oral and written test, questionnaire.	Students in the flipped class outperformed those in the traditional class students in speaking.

## Data Availability

All data generated by this review study are present in the manuscript.

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
