# Peer review of "Cognitive Gain in Digital Foreign Language Learning"

_brainsci, 2023, doi:10.3390/brainsci13071074_

Round 1

Reviewer 1 Report

Comments and Suggestions for Authors

Tha manuscript reports on a well designed and carefully conducted literature review. Some additional work to establish why this review was necessary and how it connects to existing gaps in the literature would benefit the paper.  

Author Response

Dear Reviewer,

Many thanks for your inspiring comments. We have tried to incorporate them in our revised manuscript.

Best,

Authors

Reviewer 2 Report

Comments and Suggestions for Authors

  • The authors are suggested to reorganize the abstract. In the current edition, the results/findings are presented twice. Is one of them referring to the general findings in the literature and the other referring to the specific findings based on the systematic review?

  • The authors are kindly suggested to revise the way they presented the research gap. After reading the introduction, it seems to be the case that the answers to the two research questions are readily available. For instance, beside the Finnish study, it seems to be the case that digital language learning in general could contribute to cognitive gains in foreign language learning.

  • The authors are suggested to include the non-open access articles in the review as well because inter-library service is quite convenient in Czech Republic. Also, with the inclusion of those non-open access articles, the complete picture could be better revealed.

  • There are several databases (indeice) provided by Web of Science. The authors are kindly suggested to specify which indices they chose.

  • The authors simply mentioned that “the research team carefully evaluated each one (article)” and “nine studies were identified that perfectly met all the criteria”. How exactly was the process? Did those 256 articles screened by two independent (sets of) scholars first? What was the initial number of the overlapping articles? How were discrepancies resolved?

  • The full name of LMS was not provided.

  • What about studies that do not fall into the two general important outcomes the authors mentioned on p. 4? For instance, what do the results from Liao et al. (2020) inform us about the two research questions the authors asked?

  • In Discussion, the authors mentioned that the answer to the first research question is not conclusive because “digital foreign language learning has its cons and pros depending on a number of variables, such as a method used for learning a foreign language, students´ learning needs, motivation or a digital tool employed.” However, the fact that different studies might not be comparable (because there were always different variables in different studies)” are something we all know in advance. The authors are suggested to reorganize the information package and try to highlight the importance of the techniques/strategies earlier.

Author Response

(The authors gave the same response as above.)

Reviewer 3 Report

Comments and Suggestions for Authors

The systematic is very well conducted and results are sound. I would suggest just a minor improvement. In the section "materials and methods", the paper may benefit from the inclusion of a flow chart where authors depict the search strategy, total records, records included, duplicates removed, papers excluded (and to which exclusion criteria they are related to), final paper selection, etc.  

Author Response

(The authors gave the same response as above.)
